# `NOFLITE`: Learning to Predict Individual Treatment Effect Distributions

**Toon Vanderschueren** *toon.vanderschueren@kuleuven.be*
*KU Leuven*
*University of Antwerp*

**Jeroen Berrevoets** *jb2384@cam.ac.uk*
*University of Cambridge*

**Wouter Verbeke** *wouter.verbeke@kuleuven.be*
*KU Leuven*

**Reviewed on OpenReview:** *https://openreview.net/forum?id=EjqopDxLbG*

## Abstract

Estimating the effect of a treatment on an individual's outcome of interest is an important challenge in various fields, such as healthcare, economics, marketing, and education. Previous work in machine learning has focused on estimating the expected value of the treatment effect. However, effective personalized decision-making requires more than just the treatment expected effect; it requires knowing the entire treatment effect distribution. Knowing this distribution allows analyzing the treatment's expected utility or quantifying the uncertainty regarding a treatment's effect. This information is essential for prescribing optimal treatments. The ability of a model to predict accurate individual treatment effect distributions is captured by its likelihood. In light of this, we propose a novel neural architecture, `NOFLITE`, that uses normalizing flows to directly optimize this likelihood, while simultaneously learning flexible estimates of the individual treatment effect distribution. Experiments on various semi-synthetic data sets show that `NOFLITE` outperforms existing methods in terms of loglikelihood. Moreover, we illustrate how the predicted distributions can enable an in-depth analysis of the treatment effect and more accurate decision-making.

## 1 Introduction

Knowing how a certain treatment or action will affect an instance's outcome of interest is of great importance in various domains, such as healthcare (Berrevoets et al., 2020), marketing (Devriendt et al., 2021), education (Olaya et al., 2020), and economics (Vanderschueren et al., 2023a). A wide variety of existing work has looked at using machine learning (ML) for estimating the individual treatment effect, to help decision-makers optimize treatment assignment at an individual level. Existing work on treatment effect estimation in ML has proposed novel approaches based on a variety of different ML algorithms, including neural networks (Johansson et al., 2016; Shalit et al., 2017; Zhang et al., 2020; Curth & van der Schaar, 2021a), Gaussian processes (Alaa & van der Schaar, 2017), and decision trees (Hill, 2011; Rzepakowski & Jaroszewicz, 2012; Wager & Athey, 2018), as well as general meta-learners (Künzel et al., 2019; Curth & van der Schaar, 2021b).

In spite of this growing body of literature, existing work has almost exclusively focused on accurately estimating the *expected* value of the treatment effect. We argue, however, that a more comprehensive approach is needed: *to effectively support decision-making, we require accurately modeling the effect's entire distribution*. Such an approach is essential for adopting ML for treatment decision-making in practice. First, it unlocks a wide range of descriptive statistics, enabling **a more detailed analysis** of the treatment effect. For example, it allows for reasoning about uncertainty of events resulting from a treatment: e.g., to get uncerainty

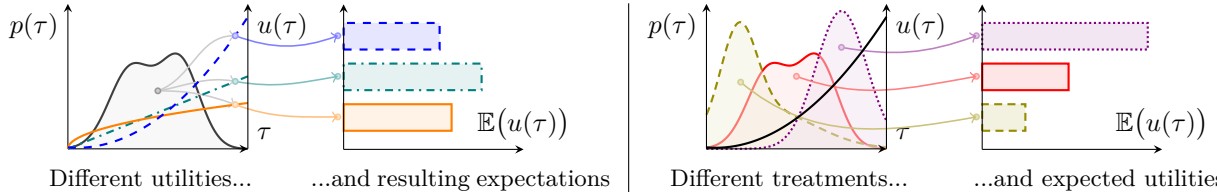

Figure 1: **Optimizing treatment decisions requires knowing the treatment effect distribution.** Predicting the treatment effect distribution $p(\tau)$ allows for assessing a treatment's utility by pairing it with a utility function $u(\tau)$ and obtaining its expected utility $\mathbb{E}\big(u(\tau)\big)$. Consequently, the individual treatment effect distribution is instrumental to analyzing treatment utilities (on the left) and comparing treatment preferences (on the right). *Left:* For a given treatment and the corresponding treatment effect distribution $p(\tau)$, we can compare different utility functions $u(\tau)$ and their expected utilities. *Right:* Similarly, for a given utility function $u(\tau)$, different treatment effect distributions $p(\tau)$ incur different expected utilities. These types of analyses are not possible using only the individual's expected treatment effect $\mathbb{E}(\tau)$, illustrating the importance of estimating the entire treatment effect distribution $p(\tau)$ for personalized decision-making.

intervals or estimate the probability of the treatment's effect being positive. This way, the treatment effect distribution subsumes other common estimands which focus on a single attribute of this distribution, such as the mean, median, or other quantiles. Second, the treatment effect distribution is essential for **deciding upon the optimal treatment**. By pairing the treatment effect distribution with a utility function, we can obtain the expected utility (see Figure 1 for a graphical illustration). Utility functions have been influential in a variety of fields, including economics (Hall, 1920; Kahneman & Tversky, 1979), game theory (Nash Jr, 1950), insurance (Spence & Zeckhauser, 1978), design (Thurston, 1991), and healthcare (Pliskin et al., 1980).

**Contributions.** The goal in this work is to estimate individual treatment effect distributions. To obtain good estimates of this distribution, we require learning a model with a high *likelihood* from observational data. This work addresses treatment effect estimation through this lens and, in doing so, makes three contributions. **1.** To learn models with high likelihood, we propose a novel neural architecture, NOFLITE, which employs normalizing flows to learn flexible estimates of the treatment effect distributions. **2.** We propose an end-to-end training strategy to directly maximize the metric of interest–the model's likelihood– while regularizing to account for treatment assignment bias. This way, we aim to learn an unbiased model with high likelihood given the (unknown) test distribution from observational training data. **3.** We evaluate our method empirically and compare performance to other state-of-the-art approaches using three semi-synthetic data sets. Contrary to existing work, this evaluation is centered around the model's likelihood.

## 2 Related work

Our work builds upon existing work in (individual) treatment effect estimation and normalizing flows. In this section, we discuss the most closely related literature for each category in turn.

### 2.1 Treatment effect estimation

A wide variety of methodologies have been proposed for estimating the *expected* individualized treatment effect (ITE). For a complete overview, we refer to Yao et al. (2021). Even though estimating *the distribution* of the ITE has received far less attention, several methods that were originally proposed for estimating the *expected value* are nevertheless capable of learning distributional estimands. These methods include Bayesian Additive Regression Trees (BART) (Hill, 2011) and Causal Multi-task Gaussian Processes (CMGP) (Alaa & van der Schaar, 2017), as well as a variety of metalearners – general combinations of supervised ML methods (Künzel et al., 2019). Particularly relevant to our work are recent approaches based on generative neural networks. On the one hand, generative adversarial networks have been used to learn the counterfactual distribution and deal with confounding bias, for binary treatments (GANITE; Yoon et al., 2018) and

| Method | Neural? | Likelihood? | Confounding? | Reference |
|--------|:-------:|:-----------:|:------------:|-----------|
| BART | ✗ | ✓ | ✗ | Hill (2011) |
| CMGP | ✗ | ✓ | ✓ | Alaa & van der Schaar (2017) |
| CEVAE | ✓ | ✗ | ✗ | Louizos et al. (2017) |
| GANITE | ✓ | ✗ | ✓ | Yoon et al. (2018) |
| CCN | ✓ | ✗ | ✓ | Zhou et al. (2022) |
| NOFLITE | ✓ | ✓ | ✓ | (Ours) |

Table 1: **Literature table.** We compare NOFLITE with existing methodologies for estimating individual treatment effect distributions. Three dimensions are considered: **(1)** whether a neural network is used, **(2)** whether the model's likelihood $\mathcal{L}_\theta$ is optimized directly, and **(3)** whether it adjust for confounding.

multiple treatments with continuous dosages (SCIGAN; Bica et al., 2020). On the other hand, variational autoencoders have been used to adjust for a hidden confounder using a (noisy) proxy (Louizos et al., 2017).

To the best of our knowledge, the only work explicitly looking at learning individual treatment effect distributions is the recently proposed Collaborating Causal Networks (CCN) (Zhou et al., 2022). As opposed to NOFLITE, CCN does not learn by maximizing the model's likelihood, but rather uses collaborating networks (Zhou et al., 2021), where one network is trained to estimate the cumulative distribution function and another model estimates its inverse. An additional, more subtle distinction is that they focus on estimating and evaluating the distribution of the potential outcomes, instead of the treatment effect distribution. A structured comparison of existing methodologies and the proposed NOFLITE is provided in Table 1.

In this work, we rely on a generative neural network. More specifically, we estimate the treatment effect distribution using normalizing flows, a flexible type of generative model that transforms a distribution to a simple prior through a series of learned transformations. Similar in spirit to our work is the recent Interventional Normalizing Flows (Melnychuk et al., 2022). They propose to use normalizing flows to estimate the density of the *average* treatment effect resulting from an intervention, building upon the theoretical results of Kennedy et al. (2023). In contrast, we aim to predict distributions at an *individual* level.

Although our work focuses on the static setting, other work has looked at forecasting treatment outcomes over time (Bica et al., 2019; Vanderschueren et al., 2023b). In this setting, CF-ODE has recently been proposed for learning uncertainty estimates for treatment outcomes over time (De Brouwer et al., 2022).

The fact that the expectation of the treatment effect is insufficient for decision-making has motivated previous work in related areas. Most closely related to ours are methods for predicting the treatment effect distribution at the population level, instead of the individual level (DiNardo et al., 1995; Chernozhukov et al., 2013; Firpo & Pinto, 2016; Kennedy et al., 2023; Melnychuk et al., 2022). Other work has been done on different, but related problems, such as estimating quantile treatment effects (Chernozhukov & Hansen, 2005; Firpo, 2007), finding the treatment regime that optimizes the quantile treatment effect (Wang et al., 2018), conformal inference of treatment effects (Lei & Candès, 2021; Alaa et al., 2023), or bounding the potential outcomes to support decision-making (Makar et al., 2020). More generally, other related work in machine learning aims to estimate confidence intervals or distribution properties for more comprehensive off-policy evaluation (e.g. Jiang & Huang, 2020; Chandak et al., 2021b;a; Xu et al., 2022; Zhang et al., 2023; Wu et al., 2023) or learns to optimize the distribution of returns of a reinforcement learning agent (Bellemare et al., 2017; 2023).

## 2.2 Normalizing flows

Our approach builds upon a type of deep generative model called normalizing flows. Normalizing flows offer distinct advantages over other types of generative models like generative adversarial networks (Goodfellow et al., 2014) or variational autoencoders (Kingma & Welling, 2014). The key benefit of using normalizing flows is that they allow for an exact evaluation of the density. Consequently, the model can be directly

optimized for the metric we are interested in: the model's likelihood. We provide a brief introduction to normalizing flows below. For a more detailed overview, we refer to Papamakarios et al. (2021).

A normalizing flow is an invertible mapping $\mathbf{g} : \mathcal{Y} \mapsto \mathcal{Z}$ from the empirical/original data space $\mathcal{Y}$ to a latent space $\mathcal{Z}$ (Rezende & Mohamed, 2015; Dinh et al., 2017). During *training*, the flow learns to map the empirical distribution $p(y)$ to a known (simple) prior distribution $p(z)$, typically a Gaussian distribution. The mapping $\mathbf{g}$ consists of a series of invertible transformations $\mathbf{g}(y) = g_1 \circ \cdots \circ g_k(y)$ with parameters $\theta$ learned by a neural network. This way, the density can be obtained using the change of variables formula:

$$p(y) = p_Z\big(\mathbf{g}(y)\big) \left| \det \left( \frac{\partial \mathbf{g}(y)}{\partial y} \right) \right|. \tag{1}$$

Using this formulation, we can evaluate the model's density exactly. Consequently, we can optimize the mapping $\mathbf{g}$ to directly maximize the model's likelihood. After training, *inference* can be done by sampling from the simple prior $p(z)$ and transforming the sample based on the inverse flow $\mathbf{g}^{-1}(z)$.

Normalizing flows have been successfully applied in a wide range of tasks, such as generating images (Dinh et al., 2014; 2017; Kingma & Dhariwal, 2018), audio (Oord et al., 2018; Kim et al., 2018; Tran et al., 2019; Prenger et al., 2019), and graphs (Madhawa et al., 2019), as well as reinforcement learning (Haarnoja et al., 2018; Ward et al., 2019; Schroecker et al., 2019) and energy forecasting (Dumas et al., 2022; Ge et al., 2020).

Different families of transformations $g$ have been proposed in the literature on normalizing flows. In this work, we use deep sigmoidal flows (Huang et al., 2018). Our choice is motivated by sigmoidal flows offering a flexible transformation – the resulting flow is a universal approximator – and their excellent empirical performance. Deep sigmoidal flows use a transformation $g$ that is a strictly monotonic neural network. The parameters of this network are given by a conditioner network. To comply with the monotinicity requirement, the learned transformer parameters are restricted to strictly positive weights and strictly monotonic activation functions–more specifically, a sigmoid activation. Although the inverse transformation $g^{-1}$ is not known analytically, it can be approximated numerically. There is a wide variety of other flow transformations that could potentially also be used with NOFLITE, including gaussianizations flows (Chen & Gopinath, 2000; Meng et al., 2020), residual flows (Chen et al., 2019), or neural spline flows (Durkan et al., 2019).

## 3 Problem Formulation

**Notation.** We describe our problem setting using the Neyman-Rubin potential outcomes framework (Rubin, 2005). Let each instance be defined by covariates $X \in \mathcal{X} \subset \mathbb{R}^d$, a binary treatment indicator $T \in \mathcal{T} = \{0, 1\}$, and an outcome $Y \in \mathcal{Y} \subset \mathbb{R}$. Let the potential outcomes $Y^{(0)}, Y^{(1)} \in \mathcal{Y} \subset \mathbb{R}$ be defined as the outcomes that would be observed given treatment $T = 0$ and $T = 1$.

**Goal.** Given an individual's covariates $x$, we are interested in predicting its individual treatment effect[1] $p(\tau)$, i.e., the difference between both potential outcomes $\tau = Y^{(1)} - Y^{(0)}$. Most existing work aims to learn an individual's *expected* treatment effect:

$$\mathbb{E}(\tau) = \mathbb{E}\big(Y^{(1)} - Y^{(0)} \mid X = x\big). \tag{2}$$

We argue, however, that this expectation itself is insufficient for decision-making in many applications. Instead, we need to learn the *distribution* of the individual treatment effect:

$$p(\tau) = P\big(Y^{(1)} - Y^{(0)} \mid X = x\big) = P\big(Y^{(1)} \mid X = x\big) - P\big(Y^{(0)} \mid X = x\big). \tag{3}$$

This distribution can be used to optimize an individual's treatment. For example, a decision-maker can use it to assess probabilistic statements, such as the probability of the treatment effect being strictly positive $p(\tau > 0)$. Alternatively, the ITE distribution can be used to evaluate treatment decisions by pairing it with a (personalized) utility function to obtain the expected utility: $\mathbb{E}\big(u(\tau)\big) = \int u(\tau)p(\tau) \, d\tau$, see Figure 1.

---

[1]When using the term Individual Treatment Effect (ITE), we refer to the instance's measured covariates included in $X$. Note, however, that these covariates need not completely describe this individual and, because of this, may refer to multiple individuals. The only requirement is that $X$ satisfies the ignorability assumptions (1-3). Because of this distinction, earlier work has argued that it is more precise to denote $\tau$ as the Conditional Average Treatment Effect (CATE), see Vegetabile (2021).

Our goal is to obtain good estimates of the treatment effect distribution, i.e., to obtain a model $\theta \in \Theta$ with a *high likelihood* $p(\tau|\theta) = \Pi_{i=1}^{n} p(\tau_i|\theta) = \Pi_{i=1}^{n} p(y_i^{(1)} - y_i^{(0)}|\theta)$ given a (hypothetical) test set $\mathcal{D}_{\text{test}} = \{(x_i, y_i^{(0)}, y_i^{(1)})\}_{i=1}^{n}$ containing both counterfactuals. Compared to point estimates such as the mean squared error, the likelihood can incorporate uncertainty and capture the entire data distribution. This way, it provides a more comprehensive measure of model performance and facilitates more robust decision-making.

**Data and assumptions.** We assume access to an observational dataset $\mathcal{D}_{\text{train}} = \{(x_i, t_i, y_i^{(t_i)})\}_{j=1}^{n}$ sampled from the joint distribution $p(X, T, Y)$. Learning a model for estimating the individual treatment effect distribution from this data is challenging for several reasons. We only observe one factual outcome $Y^{(t)}$ in practice, while the other, counterfactual outcome $Y^{(1-t)}$ is never observed. Consequently, the treatment effect itself is never observed, which is known as the fundamental problem of causal inference (Holland, 1986). Additionally, because the data are observational, treatments were assigned by a (potentially unknown) policy based on instance covariates. Therefore, it is necessary to adjust for confounding in order to obtain unbiased estimates of $p(y^{(t)})$ and, consequently, $p(\tau)$. A final challenge is that we need to learn each individual's entire distribution based on only one sample for each individual by leveraging data from similar individuals.

To identify the individualized treatment effect from observational data, we require the following standard assumptions (Rosenbaum & Rubin, 1983; Rubin, 2005):

**Assumption 1 (Consistency)** *An instance's observed outcome given a treatment is equal to its potential outcome:* $Y|X, T = Y^{(t)}|X$.

**Assumption 2 (Overlap)** *Each instance has a strictly positive probability of receiving each treatment:* $0 < \mathbb{P}(T = t \mid X = x) < 1$, $\forall\, t \in T$, $\forall\, x \in \mathcal{X}$.

**Assumption 3 (Unconfoundedness)** *Potential outcomes are independent of the treatment given the covariates:* $(Y^{(0)}, Y^{(1)}) \perp\!\!\!\perp T \mid X$.

## 4 NOFLITE: Estimating ITE distributions using normalizing flows

To tackle the problem formulated above, we propose NOFLITE[2]: a neural architecture using normalizing flows for estimating individual treatment effects distributions. A high-level overview of the architecture is shown in Figure 2. NOFLITE consists of two parts. The first part learns to predict a simple, conditional prior $p(z|x, t)$, in this case a Gaussian distribution parametrized by $(\mu, \sigma)$. The second part learns to transform this prior to a more complex posterior distribution of the potential outcome $p(y|x, t)$. The entire model is trained end-to-end by directly maximizing its likelihood, while regularizing to deal with confounding. Both the model architecture and training procedure are explained in more detail below.

### 4.1 Architecture

NOFLITE's architecture consists of two parts (see Figure 2): (1) an encoder $f$, i.e., a neural network that estimates a simple, parametrized prior distribution, and (2) a normalizing flow $\mathbf{g}^{-1}$ to transform this prior to a more complex posterior distribution. In the following, we provide a detailed description of both modules.

#### 4.1.1 Encoder: learning conditional priors

The first part of the model is a neural network that encodes the input $(x, t)$ as a simple prior distribution $p(z|x, t)$. This prior can be seen as a first approximation of the empirical distribution. We use a normal distribution defined by parameters $\mu$ and $\sigma$: $f : (X, T) \mapsto (M, \Sigma)$. Depending on the application, other distributions could be used, such as a uniform or log-normal distribution. The only requirements are that the distribution is defined by a finite set of parameters and that its likelihood can be computed analytically.

---

[2] All code is available at https://github.com/toonvds/NOFLITE.

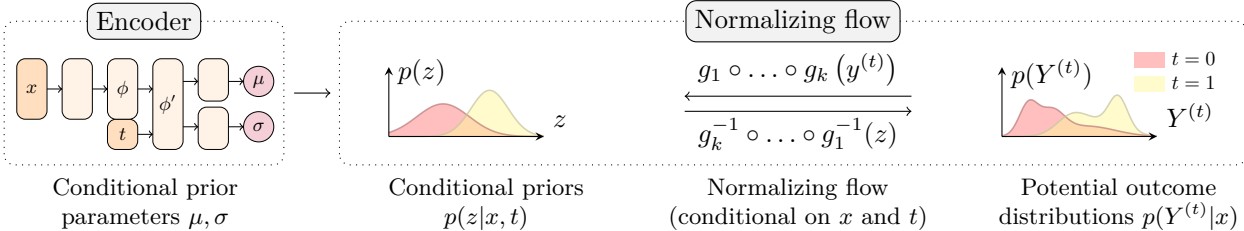

Figure 2: **NOFLITE architecture overview.** We visualize the S-learner configuration of our method. *Optimization:* NOFLITE learns (1) a conditional prior $p(z|x,t)$ for each instance, and (2) an invertible mapping $\mathbf{g}(y)$ from that conditional prior to the empirical distribution $p(y^{(t)}|x)$, possibly conditioned on $x$ and/or $t$. Both the encoder and normalizing flow use neural networks and are trained with gradient descent. *Inference:* Based on input $(x,t)$, the conditional prior $p(z|x,t)$ is estimated. Samples $z$ are drawn from this prior and transformed using the inverse flow $\mathbf{g}^{-1}(z)$ to estimate the distribution of each potential outcome $p(y^{(t)}|x)$.

To deal with confounding, $x$ is first transformed to a balanced representation $\phi$. This is achieved by learning a representation in which the distributional distance between the treated and non-treated representations is minimized (Johansson et al., 2016; Shalit et al., 2017). The training procedure is discussed in detail below.

### 4.1.2 Normalizing flow: learning complex posteriors

Although the encoder could be trained on its own to maximize the likelihood of the conditional prior, the model's hypothesis space would in that case be limited to simple parametric distributions, such as the normal distribution. Therefore, to augment the representational capacity of our model, the second part of the network uses a normalizing flow that gradually transforms the simple prior distribution $p(z|t,x)$ into a more complex posterior distribution $p(y^{(t)}|x,t)$ that can match more complex potential outcome distributions. The number of flow transformations $k$ in $\mathbf{g}$ is a hyperparameter that can be tuned. This number can be used to tune the complexity of the transformation and, therefore, the complexity of the posterior distribution.

Normalizing flows are an active field of research and, as such, new flow transforms continue to be proposed. In principle, any type of normalizing flow is compatible with NOFLITE, as long as some requirements are fulfilled. First, they have to be compatible with a univariate distribution (e.g. coupling layers (Dinh et al., 2017) are not suitable). Second, we have to be able to compute both $\mathbf{g}$ and $\mathbf{g}^{-1}$. Nevertheless, these requirements leave a variety of flow transformations (e.g., Durkan et al., 2019; Chen et al., 2019), with several potential candidates that have been shown to be universal approximators (e.g., Huang et al., 2018; Meng et al., 2020).

In this work, we instantiate $\mathbf{g}$ with a sigmoidal flow (Huang et al., 2018). This flow type offers great flexibility and is a universal approximator. The transformation is defined by a monotonic transformer network, whose parameters are estimated by a second conditioner network. By constraining the outputs of the conditioner network, the monotonicity of the transformer network is ensured. Specifically, the transformer network has strictly positive weights and strictly monotonic activation functions. The conditioner network can take different factors into account, such as covariates $x$ and/or treatment $t$. Although the inverse mapping is not known analytically, we can use an interpolation search to find it numerically.

### 4.1.3 Metalearner configuration

We implement our model in two metalearner configurations. For the *S-learner* configuration, the encoder concatenates the balanced representation $\phi$ with the treatment $t$ and uses this combination as inputs to predict the parameters of the conditional prior $p(z|x,t)$. For the *T-learner* configuration, both treatments have individual output heads after the shared balanced representation $\phi$. Similarly, we define two corresponding metalearners for the normalizing flow. The *S-learner* shares the flow across treatments, potentially conditioning the flow transformation on the treatment–depending on the flow type. The *T-learner* learns a separate flow per treatment. Either one might be more suitable, depending on the data generating process; we see this is a hyperparameter that can be tuned.

## 4.2   Optimization

The model is trained end-to-end to simultaneously estimate accurate treatment effect distributions by maximizing the likelihood, while regularizing to deal with confounding:

$$\mathcal{L}_{\text{NOFLITE}} = -\mathcal{L}_{\text{LL}} + \lambda \mathcal{L}_{\text{MMD}} \tag{4}$$

with hyperparameter $\lambda \in \mathbb{R}^+$ trading-off likelihood maximization and balancing.

To calculate the *model likelihood*, the encoder and normalizing flow cooperate and meet in the middle, i.e., in the latent space $Z$. On the one hand, the encoder maps the inputs $(x, t)$ to a prior distribution $\mathcal{N}(\mu, \sigma)$. On the other hand, the normalizing flow $g$ maps the outcome $y$ to a latent $z$. The goal of both components is to cooperate and maximize the likelihood of $z$ given $\mathcal{N}(\mu, \sigma)$. This results in the following training objective:

$$\mathcal{L}_{\text{LL}} = \log p_Z\big(\mathbf{g}(y)\big) + \log \left| \det \left( \tfrac{\partial \mathbf{g}(y)}{\partial x} \right) \right|. \tag{5}$$

The first term, the likelihood of $\mathbf{g}(y)$, can be computed analytically given the Gaussian prior from the encoder. The second term, the log determinant of the Jacobian, can be calculated from the flow transformation. This way, using a normalizing flow allows computing the likelihood exactly and, therefore, optimizing it directly. In practice, we minimize the negative log-likelihood $-\mathcal{L}_{LL}$, which is equivalent to maximizing this log-likelihood.

We learn a *balanced representation* to deal with confounding (Johansson et al., 2016; Shalit et al., 2017). This is achieved by minimizing the distributional distance between the representations of different treatments. Specifically, we use the linear Maximum Mean Discrepancy (MMD) (Gretton et al., 2012):

$$\mathcal{L}_{\text{MMD}} = 2 \left\| \frac{1}{n_0} \sum_{i \in \mathcal{D}_0} \phi(x_i) + \frac{1}{n_1} \sum_{j \in \mathcal{D}_1} \phi(x_j) \right\|^2, \tag{6}$$

where $\mathcal{D}_0$ and $\mathcal{D}_1$ denote the control and treatment group with $n_0$ and $n_1$ elements (per batch), respectively.

## 4.3   Inference

Inference is done in two steps. For each instance, samples are drawn from its prior distribution $z \sim N\big(\mu(x, t), \sigma(x, t)\big)$. These samples are then transformed to samples from the posterior using the inverse mapping: $\hat{y} = \mathbf{g}^{-1}(z)$. After this process is completed for both treatments, a sample for the ITE is obtained by taking the difference between each sampled potential outcome: $\hat{\tau} = \hat{y}^{(1)} - \hat{y}^{(0)}$.

# 5   Results

In this section, we evaluate our proposed approach and compare it with several benchmarks. Our main goal is to assess whether NOFLITE learns to predict accurate individual treatment effect distributions from different types of observational data sets. More specifically, our experiments aim to answer three questions. *(1) Does NOFLITE predict accurate individual treatment effect distributions?* This is our main question of interest. We evaluate this with the loglikelihood. *(2) Does the predicted distribution allow for a more detailed analysis of individual the treatment effect, based on statistics derived from this distribution?* This is evaluated by looking at the predicted distribution's expected value (using the precision in estimation of heterogeneous treatment effects), as well as the derived confidence intervals (using the intersection-over-union and empirical coverage). *(3) Does the predicted distribution enable accurate decision-making when paired with a utility function?* We evaluate this by looking at the accuracy of the recommended treatments.

The remainder of this section starts by describing our experimental setup, including data sets and benchmark methodologies in Section 5.1 and evaluation metrics in Section 5.2. The empirical results are presented in Section 5.3. More information on hyperparameter tuning for NOFLITE is provided in Appendix A.2.

## 5.1 Data and benchmarks

Evaluating individualized estimates of the treatment effect is challenging, because only one outcome is observed and, because of that, we do not have access to the ground truth. To overcome this challenge, we follow existing work and evaluate based on semi-synthetic data. More specifically, in this work, we evaluate NOFLITE using commonly used benchmark data sets: IHDP, EDU, and News. These are introduced briefly in the following; Appendix A.1 provides more detailed information on the data generating processes.

**IHDP.** The Infant Health and Development Program (IHDP; Hill, 2011) is a semi-synthetic data set that is commonly used to evaluate machine learning models for causal inference. Covariates are based on a real-world randomized experiment in which some infants ($n = 747$) were targeted with child care and home visits. The resulting outcome is simulated based on these covariates ($x \in \mathbb{R}^{25}$) and the treatment, with the resulting treatment effect following a normal distribution. Although this dataset has recently been criticized (Curth et al., 2021), we include it due to it being a widely-used benchmark.

**EDU.** The Education data set (EDU; Zhou et al., 2022) measures the effect of providing a mother with adult education benefits on their children's learning. The data is simulated based on covariates $x \in \mathbb{R}^{32}$ using two (non-linear) neural networks, with added Gaussian ($t = 0$) or exponential noise ($t = 1$). For both potential outcomes, the noise level depends on one of the covariates; more specifically, a single binary variable. Confounding is introduced through covariate-based propensities and removing well-balanced subjects.

**News.** The News data set (Johansson et al., 2016) shows the effect of reading an article on either mobile or desktop ($t$) on the reader's experience ($y$), based on the article's content in word counts ($x$). The data is simulated using a topic model $z(x)$, which is used to define two centroids in the topic space: $z_0^c$ (desktop) and $z_1^c$ (mobile). Device assignment and reader experience are both simulated based on the similarity of the article's topic $z(x)$ to the centroids $z_0^c$ and $z_1^c$. The resulting data is very high-dimensional ($d = 3{,}477$).

**Benchmark methodologies.** We compare NOFLITE with several other machine learning models capable of learning distributions. First, we compare against Causal Multi-task Gaussian Processes[3] (CMGP; Alaa & van der Schaar, 2017). Second, we compare against other methods relying on generative neural networks. The Causal Effect Variational Autoencoder (CEVAE; Louizos et al., 2017) uses a variational autoencoder to model latent confounders given noisy proxies. GANITE (Yoon et al., 2018) uses generative adversarial networks to deal with selection bias and to obtain a probabilistic estimate of the treatment effect. Finally, we compare against Causal Collaborating Networks (CCN; Zhou et al., 2022).

## 5.2 Performance metrics

The main goal in this work is to predict accurate individual treatment effect distributions. Therefore, the main metric of interest is the model's *loglikelihood*, which allows for a comprehensive assessment of each instance's predicted distribution, by quantifying how likely the test data is given the predicted distributions. We estimate the loglikelihood as follows. For each instance $i$, we sample 500 samples from the model $\theta$ based on the covariates $x_i$. Then, for each instance, we fit a Gaussian kernel density estimator $\mathtt{kde}_i(x_i, \theta)$ using these samples and estimate the loglikelihood of the true treatment effect according to this kernel density estimator $\log p(\tau_i \,|\, \mathtt{kde}_i(x_i, \theta))$. The loglikelihood is averaged over all instances:

$$\mathrm{LL} = \frac{1}{n} \sum_{i=1}^{n} \log p\big(\tau_i \,|\, \mathtt{kde}_i(x_i, \theta)\big). \tag{7}$$

The loglikelihood evaluates the predicted distributions globally. As argued in the motivation, statistics derived from the distribution can be used to facilitate decision-making. Therefore, to facilitate a more holistic assessment of performance, we present additional metrics that analyze specific properties of the

---

[3]For the high dimensional News data set, we take the first 100 principal components to avoid excessive training times. This number was tuned based on a validation set.

predicted distribution. First, we use the square root of the *precision in estimation of heterogeneous effects* ($\sqrt{\text{PEHE}}$, see Hill, 2011) to evaluate the accuracy of the expected value of the individual treatment effect:

$$\text{PEHE} = \frac{1}{n} \sum_{i=1}^{n} \left( \left( y_i^{(1)} - y_i^{(0)} \right) - \mathbb{E} \left( \hat{y}_i^{(1)} - \hat{y}_i^{(0)} \right) \right)^2 = \frac{1}{n} \sum_{i=1}^{n} (\tau_i - \hat{\tau}_i)^2. \tag{8}$$

Moreover, we evaluate the *empirical coverage* (Cov) of the estimated 90% confidence interval $\hat{CI}$, i.e., the probability that an observed sample of the ITE lies within the estimated interval:

$$\text{Cov} = \frac{1}{n} \sum_{i=1}^{n} \mathbb{1}(\tau_i \in \hat{CI}_i). \tag{9}$$

For the 90% CI, the empirical coverage should ideally be 0.90 with small variance between iterations. For some data sets, we know the distribution that was used to generate the data. In these cases, we can additionally compare the predicted and true confidence intervals. We propose to evaluate their overlap using the *intersection-over-union* (IoU) comparing the predicted and ground truth confidence intervals:

$$\text{IoU} = \frac{CI \cap \widehat{CI}}{CI \cup \widehat{CI}}. \tag{10}$$

This metric is bounded between 0 and 1. The worst value, 0, indicates an empty intersection. The best value, 1, indicates an intersection equal to the union and, consequently, an estimated confidence interval that is equal to the ground truth.

Finally, we evaluate the quality of decisions made based on the predicted distributions, for a given utility distribution. The decision to treat or not is based on whether the expected utility is positive. For IHDP, we use $u(\tau) = (\tau - 4)^3$, given that the average treatment effect will be 4 on average (Hill, 2011). For EDU, we use $u(\tau) = (\tau - 1)^3$, as the treatment effect was empirically observed to be approximately 1 on average. The predicted optimal treatment decision is compared with the theoretical optimal based on the ground truth distribution using the *accuracy* (Acc). For the News data, we cannot do this analysis, as the true distribution is not known. For all metrics and models, we use 500 samples (per instance) for evaluation.

## 5.3  Empirical results

We compare the different models for the IHDP, EDU, and News data sets in Table 2. In terms of loglikelihood, NOFLITE obtains the best performance out of all methods under consideration for each data set. These findings demonstrate NOFLITE's ability to learn accurate individual treatment effect distributions from a variety of observational data sets and associated data generating processes. For the IHDP data set, NOFLITE slightly outperforms the next-best model, CMGP. Both methods use a Gaussian prior, which matches IHDP's data generating process. On the EDU data set, the data generating process is more complex and requires more flexiblity. Indeed, the Gaussian prior of CMGP results in relatively worse performance and the more complex models, such as NOFLITE and CCN, perform better. This illustrates how the normalizing flows can allow for a more flexible model of posterior when required. Finally, the good performance on the News data set illustrates NOFLITE's ability of learning from high dimensional data. As opposed to CMGP, which needs PCA preprocessing, it can also handle this high dimensional data out-of-the-box.

When we look at the metrics evaluating statistics derived from the treatment effect distribution, i.e. the PEHE and IoU, there is no clear winner overall: either CMGP, CCN or NOFLITE result in the best performance. Nevertheless, for all metrics, NOFLITE's performance is competitive with the best performing methodology for each data set. This illustrates how optimizing the likelihood and learning distributions results in good performance for the metrics evaluating properties of this distribution. Nevertheless, if there is one particular metric of interest (e.g., PEHE), other objectives than NOFLITE's loglikelihood might be preferable (e.g., the mean squared error). Moreover, we observe that NOFLITE obtains relatively high accuracy for both the IHDP and EDU data sets. This illustrates that pairing the NOFLITE's predicted distributions with a utility function enables qualitative decision-making.

| | $LL$ (↑) | $\sqrt{PEHE}$ (↓) | $IoU$ (↑) | $Cov$ (90%) | $Acc$ (↑) |
|---|---|---|---|---|---|
| CMGP | $-1.95_{\pm 0.07}$ | $\mathbf{0.73_{\pm 0.09}}$ | $\mathbf{0.77_{\pm 0.01}}$ | $\mathbf{0.88_{\pm 0.00}}$ | $\mathbf{0.90_{\pm 0.01}}$ |
| BART | $-2.16_{\pm 0.05}$ | $2.22_{\pm 0.36}$ | $0.63_{\pm 0.02}$ | $0.83_{\pm 0.01}$ | $0.84_{\pm 0.01}$ |
| GANITE | — | $6.32_{\pm 0.89}$ | $0.01_{\pm 0.00}$ | $0.01_{\pm 0.00}$ | $0.55_{\pm 0.02}$ |
| CEVAE | $-2.97_{\pm 0.09}$ | $5.71_{\pm 0.89}$ | $0.25_{\pm 0.01}$ | $0.98_{\pm 0.00}$ | $0.57_{\pm 0.01}$ |
| CCN | $-2.16_{\pm 0.08}$ | $1.46_{\pm 0.17}$ | $0.66_{\pm 0.02}$ | $0.84_{\pm 0.00}$ | $0.86_{\pm 0.01}$ |
| NOFLITE | $\mathbf{-1.90_{\pm 0.01}}$ | $1.09_{\pm 0.20}$ | $0.75_{\pm 0.00}$ | $\mathbf{0.88_{\pm 0.00}}$ | $\mathbf{0.90_{\pm 0.01}}$ |

(a) **IHDP** ($n = 747; d = 25$)

| | $LL$ (↑) | $\sqrt{PEHE}$ (↓) | $IoU$ (↑) | $Cov$ (90%) | $Acc$ (↑) |
|---|---|---|---|---|---|
| CMGP | $-1.74_{\pm 0.01}$ | $\mathbf{0.22_{\pm 0.01}}$ | $0.56_{\pm 0.00}$ | $\mathbf{0.91_{\pm 0.00}}$ | $0.70_{\pm 0.00}$ |
| BART | $-1.71_{\pm 0.01}$ | $0.53_{\pm 0.01}$ | $0.56_{\pm 0.00}$ | $0.89_{\pm 0.00}$ | $0.69_{\pm 0.00}$ |
| GANITE | — | $1.26_{\pm 0.08}$ | $0.37_{\pm 0.03}$ | $0.46_{\pm 0.03}$ | $0.72_{\pm 0.01}$ |
| CEVAE | $-2.67_{\pm 0.03}$ | $2.20_{\pm 0.25}$ | $0.25_{\pm 0.00}$ | $1.00_{\pm 0.00}$ | $0.49_{\pm 0.01}$ |
| CCN | $-1.65_{\pm 0.01}$ | $0.31_{\pm 0.01}$ | $\mathbf{0.64_{\pm 0.01}}$ | $0.87_{\pm 0.00}$ | $\mathbf{0.76_{\pm 0.01}}$ |
| NOFLITE | $\mathbf{-1.62_{\pm 0.01}}$ | $0.26_{\pm 0.01}$ | $\mathbf{0.64_{\pm 0.01}}$ | $0.89_{\pm 0.00}$ | $\mathbf{0.76_{\pm 0.01}}$ |

(b) **EDU** ($n = 8,627; d = 32$)

| | $LL$ (↑) | $\sqrt{PEHE}$ (↓) | $Cov$ (90%) |
|---|---|---|---|
| CMGP | $-2.29_{\pm 0.03}$ | $2.21_{\pm 0.05}$ | $0.95_{\pm 0.00}$ |
| BART | $-2.43_{\pm 0.04}$ | $2.71_{\pm 0.12}$ | $0.97_{\pm 0.00}$ |
| GANITE | — | $18.91_{\pm 11.29}$ | $0.01_{\pm 0.00}$ |
| CEVAE | $-2.83_{\pm 0.04}$ | $3.74_{\pm 0.18}$ | $0.97_{\pm 0.00}$ |
| CCN | $-2.25_{\pm 0.03}$ | $2.23_{\pm 0.04}$ | $0.84_{\pm 0.00}$ |
| NOFLITE | $\mathbf{-2.15_{\pm 0.02}}$ | $\mathbf{2.18_{\pm 0.05}}$ | $\mathbf{0.93_{\pm 0.00}}$ |

(c) **News** ($n = 5,000; d = 3,477$)

Table 2: **Empirical results.** We compare NOFLITE against a variety of existing methods for three data sets: (a) IHDP, (b) EDU, and (c) News. For each metric, arrows indicate whether a lower (↓) or higher (↑) value is better; the ideal coverage is 90%. We show the best result in **bold**. We highlight the loglikelihood ( LL ) in gray to emphasize that this is our main metric of interest, as it evaluates the quality of the predicted distributions. For all data sets, GANITE achieves a loglikelihood of less than $-10$, indicated by a dash (—).

Finally, we illustrate how NOFLITE can be used for practical applications. We do this by training our model on an iteration of the News data set and showing the model's output for a few selected test instances, see Figure 3. Figures 3a to 3c show the ITE distribution and related statistics based on samples from the learned model. This not only allows for visualizing the estimated distribution $p(\tau_i)$ and its associated expected value $\mathbb{E}(\tau)$, but also for assessing the uncertainty using the 90% confidence interval or a boxplot. Moreover, due to the News data set being semi-synthetic, we can compare the ground truth $\tau_{\text{observed}}$ with the estimated distribution. Finally, the model can be used to consider the treatment decision for an instance by, e.g., looking at the probability of its treatment effect being strictly positive. Figure 3d shows the heterogeneity in distributions of different instances, both in terms of expected value and shape.

# 6 Conclusion

Estimating an instance's individual treatment effect distribution is an essential requirement for personalized decision-making. To this aim, we presented NOFLITE, a flexible neural method for estimating treatment effect distributions that directly optimizes the metric of interest for this task: the model's likelihood. By leveraging normalizing flows, the model is not constrained to any particular parametric distribution, but can instead trade off a simple normal distribution with a more complex posterior, depending on the data. Experiments on a variety of data sets demonstrated NOFLITE's good performance in practice and underlined its excellent representational capacity, as illustrated by its ability to obtain high likelihoods for a variety of data set sizes, dimensionalities, and data generating processes.

Future work in normalizing flows could benefit our method, as novel flow transformations will be compatible with our method and help address potential limitations of our work. First, depending on the flow transformation that is used, training or inference can be slow in normalizing flows. Second, NOFLITE introduces a variety of hyperparameters. Therefore, novel flow transformation with less hyperparameters might benefit adoption of our method in practice. Related to these points, it would be interesting to incorporate advances in other types of generative models, such as diffusion models (see e.g. Yang et al., 2022), in future work.

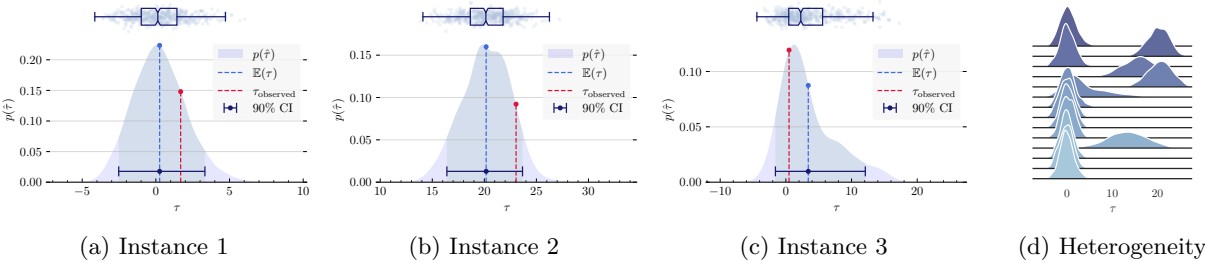

Figure 3: **NOFLITE illustration.** We visualize NOFLITE's output for selected test instances for one iteration of the News data set. *Individual treatment effect distributions:* Figures (a-c) visualize one particular instance's treatment effect distribution and related statistics based on samples from the model. *Distribution heterogeneity:* Figure (d) compares the predicted individual treatment effect distributions of several instances.

Although estimating individual treatment effect distributions can be very valuable, it constitutes a challenging problem – especially when working with high-dimensional, observational data. For many applications, the available data may be limited. Although NOFLITE can be used without flow transformations, its flexibility comes from the normalizing flows, which require data to be trained succesfully. Although NOFLITE's hyper-parameters can be tuned by looking at the fit on a validation set, validating causal inference models itself is a challenging problem (Curth & van der Schaar, 2023). Therefore, we consider it a promising area for future work to analyze NOFLITE from a theoretical perspective and come up with performance guarantees based on statistical learning theory.

Additionally, our method relies on the standard ignorability assumptions in causal inference. Judging the feasibility of these assumptions is impossible based on data alone and requires the judgment of domain experts. There is a growing body of work looking at learning treatment effects under violations of these assumptions (e.g., under hidden confounding, see Kallus et al. (2019); Oprescu et al. (2023)). Another interesting direction for future work is to analyze performance of our method in settings where the ignorability assumptions are violated, and to extend our methodology to account for these violations.

### Acknowledgments

We would like to thank Thomas Van Hout and the anonymous reviewers for insightful comments and discussions on earlier drafts of this paper. Toon Vanderschueren is supported by the Research Foundation – Flanders (FWO PhD Fellowship 11I7322N).

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

## A  Appendix

### A.1  Data sets and associated data generating processes

This section gives more extensive details on the different semi-synthetic data sets used in this work. We describe each data set and the underlying data generating process that were used to simulate it.

**IHDP** $(n = 747, d = 25;$ **Hill, 2011**).  We use data based on the standard response surface B. In this setting, potential outcomes are generated as

$$Y_i^{(0)} = \exp((x_i + A)\beta) + \varepsilon \quad \text{and} \quad Y_i^{(1)} = x_i\beta - \omega + \varepsilon \tag{11}$$

where $\varepsilon \sim \mathcal{N}(0,1)$, $W$ is a (fixed) offset matrix with each value equal to 0.5, $\beta$ is a sparse coefficient vector with each element sampled from $(0, 0.1, 0.2, 0.3, 0.4)$ with corresponding probabilities $(0.6, 0.1, 0.1, 0.1, 0.1)$. For each iteration of the data set, $\omega$ is set to ensure that the conditional average treatment effect for the treated (CATT) and conditional average treatment effect on the controls (CATC) both equal 4 on average. We use the 100 replications from github.com/clinicalml/cfrnet (Shalit et al., 2017).

**EDU** $(n = 8{,}627, d = 32;$ **Zhou et al., 2021**).  For each treatment group, a neural network $f_{y^{(t)}}$ is trained based on the observed outcomes. The potential outcomes are then simulated as

$$Y_i^{(0)} = f_{y^{(0)}}(x_i) + (2 - x_i^{23})\varepsilon_0 \quad \text{and} \quad Y_i^{(1)} = f_{y^{(1)}}(x_i) + (2 - x_i^{23})\varepsilon_1, \tag{12}$$

with $\varepsilon_0 \sim \mathcal{N}(0, 0.5^2)$ and $\varepsilon_1 \sim \exp(2)$. $x^{23}$ refers to the $23^{\text{rd}}$ covariate, which is a binary covariate indicating whether the instance's mother has received previous education.

**News** $(n = 5{,}000, d = 3{,}477;$ **Johansson et al., 2016**).  First, based on a topic model $z$, two treatment-specific centroids $z_0^c$ and $z_1^c$ are estimated in the topic space $z(x)$. Each potential outcome is then generated as the similarity between $z(x_i)$ and $z_t^c$ as

$$Y_i^{(t)} = C\big(z(x_i)^{\intercal}z_0^c + t_i \cdot z(x_i)^{\intercal}z_1^c\big) + \varepsilon,$$

with scaling factor $C$ and $\varepsilon \sim \mathcal{N}(0,1)$. Treatment assignment is modeled as $p(t_i = 1|x_i) = \frac{\exp(\kappa \cdot z(x_i)^{\intercal}z_1^c)}{\exp(\kappa \cdot z(x_i)^{\intercal}z_1^c) + \exp(\kappa \cdot z(x_i)^{\intercal}z_1^c)}$ with $\kappa = 10$.

### A.2  Hyperparameter optimization

We add more information on the chosen hyperparameters in Table 3. Given the absence of the ground truth in observational data, hyperparameter tuning was done using wandb (Biewald, 2020) using the potential outcome's loglikelihood, PEHE, and IoU of a validation set. Additionally, to tune the hyperparameter $\lambda$ for the de-biasing term, we use an inverse propensity weighted loglikelihood, allowing us to assess performance under covariate shift. Training is done using gradient descent with the Adam optimizer (Kingma & Ba, 2015). We use several types of regularization: $\ell_1$, $\ell_2$, and noise regularization (Rothfuss et al., 2019). The encoder networks uses exponential linear units (ELU) as activation functions (Clevert et al., 2015).

The optimal hyperparameters additionally give some insights into how `NOFLITE` can deal with a diversity of data generating processes. For instance, the best metalearner was the T-learner for all data sets. The

| Hyperparameter | IHDP | EDU | News |
|---|---|---|---|
| *— General —* | | | |
| Metalearner | T | T | T |
| *— Encoder —* | | | |
| Hidden layers balancer | 2 | 1 | 3 |
| Hidden layers encoder – shared | 3 | 0 | 0 |
| Hidden layers encoder – separate | 2 | 2 | 3 |
| Hidden neurons encoder | 8 | 8 | 32 |
| *— Flow —* | | | |
| Number of flow transformations $k$ | 0 | 4 | 1 |
| Flow type | — | SigmoidX | SigmoidX |
| Hidden neurons transformer | — | 4 | 2 |
| Hidden neurons conditioner | — | 16 | 32 |
| Hidden layers conditioner | — | 2 | 1 |
| *— Training settings —* | | | |
| Learning rate | 5e-4 | 5e-4 | 5e-4 |
| Batch size | 128 | 512 | 128 |
| Training steps | 5,000 | 5,000 | 10,000 |
| Regularization $\lambda_{\ell_1}$ | 1e-3 | 0 | 5e-4 |
| Regularization $\lambda_{\ell_2}$ | 5e-4 | 1e-3 | 5e-3 |
| $\lambda_{mmd}$ | 1 | 1e-2 | 1e-2 |
| Noise regularization $x$ | 0 | 0 | 1 |
| Noise regularization $y$ | 5e-1 | 1e-1 | 5e-1 |
| Truncation probability | 0 | 0 | 1e-2 |

Table 3: **Hyperparameter tuning.** We show the optimal hyperparameters for the different data sets. The flow type 'SigmoidX' refers to the deep sigmoidal flow (DSF) of Huang et al. (2018) conditioned on the balanced representation $\phi$ of $x$. During inference, drawing samples from a truncated normal distribution was observed to be more stable for the News data.

potential outcomes were generated separately for all data sets under consideration, which corresponds more closely to the T-learner's hypothesis space. Moreover, the complexity of the distribution can be tweaked depending on the data generating process: NOFLITE does not use any flows for IHDP, uses only one flow for News, and uses four flow transformations for EDU. These settings match the increasingly more complex distributions being used in the corresponding the data generating processes.

