# OpenReview forum: "NOFLITE: Learning to Predict Individual Treatment Effect Distributions"
_TMLR — Accepted by TMLR_

### Review · Reviewer_fyxB · 2023-08-10

**Summary Of Contributions:**

This paper proposes a normalizing flow based method to estimation the distributions of conditional treatment effects under standard (but strong) assumptions of no measured confounding, consistency, and overlap. They propose two meta-learner configurations for the architecture, one (S-learner) which learns one flow for both treatments and the other (T-learner) that learns a separate flow for each treatment. They present empirical results using semi-synthetic design on three datasets, Infant Health Development Program and an Education dataset, and a News dataset. Their empirical analysis investigates how their method compares to existing methods in terms of log-likelihood, PEHE, and a intersection-over-union measure of the estimated confidence intervals.


**Audience:**

Yes

**Broader Impact Concerns:**

For work that has the potential to be used in high-stakes domains and settings where evaluation is challenging, it's really important to be careful about the language used and to be transparent about the assumptions and limitations of the proposed method. As noted above, it would be important to update the paper to clarify what is meant by "individual treatment effect estimation" and to devote more discussion to the important assumptions required by the method.

**Claims And Evidence:**

Yes

**Requested Changes:**

recommendations:
- Add discussion about whether it’s reasonable to make the required assumptions and provide references to work in this space that relaxes those assumptions (e.g., Interval Estimation of Individual-Level Causal Effects Under Unobserved Confounding by Nathan Kallus, Xiaojie Mao, and Angela Zhou)
- Provide standard coverage empirical results (as described above)
- Add discussion on the difficulty of high-dimensional conditional estimation and discuss implications for proper use of the method (e.g., sample sizes required etc)
- Use more precise language like "conditional or heterogeneous treatment effect estimation" instead of "individual treatment effect estimation".


**Strengths And Weaknesses:**

Contribution:

Strengths:
+ Well-written. The main ideas are presented clearly and the paper includes nice figures/illustrations to convey key ideas.
+ Estimating the distributions of conditional treatment effects is an important but very hard problem.

problems:
- The term “Individual treatment effect” has the potential to be misleading. I think it’s better to be more precise and call these heterogeneous/conditional treatment effects or at the very least have a discussion of how the word “individual” refers to a measured covariate vector.
- Empirical results: IoU metric for evaluating the confidence intervals seems to suggest there is poor coverage although the metric is hard to evaluate. It would be more helpful to present empirical coverage results— e.g., does the 90% CI actually contain the true ITE 90% of the time.
- No theoretical guarantees. While normalizing flows have shown success in domains where evaluation is straightforward (e.g., image generation), causal inference is different since in real-world settings we never observe the ground-truth outcomes needed for evaluation. It’s not clear that the empirical results are sufficient to establish that this method will perform well in real-world contexts. Such claims would be bolstered by theoretical guarantees.

---

> ### Author Response · Authors · 2023-10-06
> **Response to Reviewer 6esJ**
>
> Dear Reviewer 6esJ,
>
> Thank you for your thoughtful comments, suggestions, and recommendations! We are very grateful for your positive feedback, appreciating the importance of this problem and the clarity of our writing. We provide point-by-point responses to the key points raised in your review below: (1) Assumptions and future work, (2) Coverage, and (3) ITE vs CATE.
>
> ___
>
> # (1) Assumptions and future work
>
> We have added a limitations section to our conclusion (see below), where we touch upon the difficulty of density estimation in general:
>
> "Although estimating individual treatment effect distributions can be very valuable, it constitutes a highly challening problem—especially when working with high-dimensional, observational data. For many applications, the available data may be limited. Although NOFLITE can be used without flow transformations, its flexibility comes from the normalizing flows, which require data to be trained succesfully. Although NOFLITE's hyperparameters can be tuned by looking at the fit on a validation set, validating causal inference models itself is a challenging problem (Curth and van der Schaar, 2023). Therefore, we consider it a promising area for future work to analyze NOFLITE from a theoretical perspective and come up with performance guarantees based on statistical learning theory."
>
> Similarly, we discuss the assumptions required for our work:
> "Additionally, our method relies on the standard ignorability assumptions in causal inference. Judging the feasibility of these assumptions is impossible based on data alone and requires the judgment of domain experts. There is a growing body of work looking at learning treatment effects under violations of these assumptions (e.g., under hidden confounding, see Kallus et al., 2019; Oprescu et al., 2023). An interesting direction for future work is to analyze performance of our method in settings where the ignorability assumptions are violated and to extend our methodology to account for these violations."
>
> Thank you for this suggestion!
>
> ___
>
> # (2) Coverage
>
> We have added empirical coverage results for the 90% confidence interval below and in the revised manuscript. We consider these results a valuable addition to our work. Thank you for this suggestion!
>
> #### **Empirical results: empirical coverage (90\% CI)**
> |              | IHDP         | EDU          | News         |
> | ------------ | ------------ | ------------ | ------------ |
> | **CMGP**     | 0.88         | 0.91         | 0.95         |
> | **BART**     | 0.81         | 0.80         | 0.98         |
> | **GANITE**   | 0.01         | 0.46         | 0.01         |
> | **CEVAE**    | 0.98         | 1.00         | 0.97         |
> | **CCN**      | 0.84         | 0.87         | 0.84         |
> | **NOFLITE**  | 0.88         | 0.89         | 0.93         |
>
> These results show that NOFLITE is able to learn useful distributions, in the sense that the derived 90\% confidence intervals are reasonably accurate. Moreover, they show how earlier approaches based on generative models either predict confidence intervals that are too narrow (GANITE) or too wide (CEVAE). Note that we also include results for BART, based on Reviewer gFHo's suggestion.
>
> We would like to add that the Intersection-over-Union (IoU)  metric can be complementary to the empirical coverage. Allow us to first again explain the IoU and its interpretation in more detail. Then, we will discuss how it differs from and complements the empirical coverage.
>
> Because we are working with synthetic data, we are in the unique situation where we know the exact confidence interval (CI). The IoU compares these true and predicted CIs. It originates from computer vision, where it is for example used to evaluate the accuracy of a segmentation map. It is calculated as the ratio of the intersection and union of the predicted and true intervals:
>
> $$\text{IoU} = \frac{CI \cap \widehat{CI}}{CI \cup \widehat{CI}}$$
>
> A value of one is achieved when the union and intersection are equal, indicating a perfect match. A value of zero indicates that the predicted and true confidence interval do not overlap at all.
>
> Why are both metrics complementary? The IoU does not take density into account, while the empirical coverage does not look at the region covered by the interval. Because of this, a model with good coverage can have a low IoU. Empirically, we indeed see differences in performance with respect to IoU and empirical coverage. For example, on the EDU data, CMGP has good coverage, but relatively worse IoU. CMGP predicts symmetric confidence intervals resulting from the normal distribution used in this method. Therefore, it cannot match the assymmetric true confidence interval. Because practitioners might also use the "shape" of the confidence interval, we believe that the IoU offers additional insights in comparing the methods.

---

> > ### Author Response · Authors · 2023-10-06
> > **Response to Reviewer 6esJ: Part 2/2**
> >
> > # (c) ITE vs CATE
> > We agree with the reviewer that it is more precise to talk about the Conditional Average Treatment Effect (CATE) instead of the Individual Treatment Effect (ITE). Thank you for pointing this out! To address this, we have add the following footnote to our revised manuscript:
> > "When using the term Individual Treatment Effect (ITE), this refers to the instance's measured covariates included in $X$. Note, however, that these covariates need not completely describe this individual and, because of this, may refer to multiple individuals. The only requirement is that $X$ satisfies the ignorability assumptions (1-3). Because of this distinction, earlier work has argued that it is more precise to denote $\tau$ as the Conditional Average Treatment Effect (CATE), see Vegetabile (2021)."
> > ___
> >
> >
> > We want to once more thank the reviewer for their time and feedback. We hope that these responses address your concerns, but please do not hesitate to ask if there are any remaining issues.
> >
> > ___
> >
> > ## _References:_
> > * Vegetabile, B. G. (2021). On the Distinction Between “Conditional Average Treatment Effects” (CATE) and “Individual Treatment Effects” (ITE) Under Ignorability Assumptions. arXiv preprint arXiv:2108.04939.
> > * Curth, A., & van der Schaar, M. (2023). In Search of Insights, Not Magic Bullets: Towards Demystification of the Model Selection Dilemma in Heterogeneous Treatment Effect Estimation. In International Conference on Machine Learning (ICML).
> > * Kallus, N., Mao, X., & Zhou, A. (2019, April). Interval estimation of individual-level causal effects under unobserved confounding. In The 22nd international conference on artificial intelligence and statistics (pp. 2281-2290). PMLR.
> > * Oprescu, M., Dorn, J., Ghoummaid, M., Jesson, A., Kallus, N., & Shalit, U. (2023). B-Learner: Quasi-Oracle Bounds on Heterogeneous Causal Effects Under Hidden Confounding. In International Conference on Machine Learning (ICML).

---

### Review · Reviewer_KX1e · 2023-09-06

**Summary Of Contributions:**

This paper introduces a method, based on Normalizing Flows, to estimate the full distribution of treatment effects. Claims are made that this full distribution enables a more accurate analysis of the treatment effect than simply estimating the expectation of that distribution. The paper optimizes the model that estimates this full distribution based on the log likelihood and is compared to recent generative approaches to estimating the treatment effect on three semi-synthetic datasets.

**Audience:**

Yes

**Broader Impact Concerns:**

None that I can ascertain.

**Claims And Evidence:**

No

**Requested Changes:**

## Major

Better clarifying details about the specific motivations of the approach.

Clarify how the log likelihood is an adequate comparative metric across the various baselines.

Inclusion of CF-ODE as an empirical baseline.

Additional analysis of the statistics enabled by the full distribution.

Analysis and discussion of how utility functions can be used to modify and/or improve the analysis of the estimated treatment effect distributions.

## Minor

More detail about how the Normalizing Flows are used in practice.

Figure 2 typo in inversion of the flow ($g_1$ is not inverted).

Reframe the metrics PEHE and IoU to discuss their use in evaluating the learned/estimated distribution.

**Strengths And Weaknesses:**

## Strengths

The use of Normalizing Flows to construct the approximate treatment effect distribution is an intriguing and reasonable idea. I feel that this contribution is a specific strength of this paper.

I agree with the position provided by the authors that estimating the full treatment effect distribution enables more informed decision making. I am encouraged to see more work that focuses on these type of efforts. I do however have some concerns (listed below) about the way this claim is presented and justified however.

The proposed NOFLITE is adequately grounded among the relevant literature. It is made clear by the authors where prior methods may not fully satisfy accurate effect estimation. I appreciated the detail provided in Section 2 about Normalizing Flows, I do feel that more could be written to explain how the inverse flow $g^{-1}(z)$ is executed. That would be a really helpful point of information to support the full conceptual grounding of the proposed modelling approach.

The construction of the NOFLITE approach is easy to understand and clearly articulated at the outset of Section 4. Figure 2 is really informative! The one aspect of this that could be improved is discussing how the encoder handles confounding. More detail is warranted there to explain the balanced representations used. (Also, the terms in Section 4.1.2 "The flow transformation uses a monotonic transformer network that is parametrized by a conditioner network..." are very jargony without any context about what they may mean. Perhaps these could be expanded upon slightly? And Section 4.1.3 seems unnecessary in the main body of the paper without supporting context in Section 2.)

## Weaknesses

There are not very clear descriptions of what the actual claims and intended contributions of this paper are, outside of the the generic framing of estimating the full treatment effect distribution with a neural architecture that harnesses normalizing flows to construct the distributional estimate. Many of the statements in the introduction are vague and imprecise, appealing to a generic intuition that the estimation of a treatment effect distribution and its presented consequences are already accepted. For example, the sentence: "By pairing the treatment effect distribution with a utility function, we can obtain the expected utility." is fine but doesn't completely convey why utility functions are important or useful nor expresses why expected utility is a desirable metric. The following sentence provides really weak justification. Appealing to the use of the metric in other fields isn't a sufficient explanation or description of why it's a good idea here. There are similar problems with how the "first" stated advantage of estimating the effect distribution is presented. What do the expanded set of descriptive statistics actually provide? Why is this useful? I think more precise statements and an expanded introduction could strengthen the paper and clearly set out the major claims being put forward by the authors.

Figure 1 would be improved by visualizing how $\mathbb{E}[p{\tau}]$ fails to adequately capture the true treatment effect. Additionally, it's unclear what the different type of utility functions may measure (Fig 1 (left)) and why they would be under consideration. This isn't sufficiently discussed in the text of the paper. For example, what does it mean to have a "personalized utility function"? Finally, Figure 1 should be situated closer to Section 3. It isn't very helpful to be placed so early (as currently presented, this may change if there were illustrations of how the expected treatment effect falls short in tandem with a more detail Introductory section).

Unfortunately, I fear that the authors missed an important comparative baseline approach in the CF-ODE of de Brouwer, et al (2022, AISTATS) which satisfies the very same considerations proposed as important dimensions of consideration in developing models for estimating treatment effect distributions (Neural, Likelihood, Confounding). The CF-ODE model satisfies all three considerations, similar to NOFLITE, and also enables inference over continuous time settings when estimating the treatment effect distributions. This is an unfortunate oversight and I would highly recommend that the authors include this method in their experimental benchmarking so the claims of NOFLITE's performance could be fully centered among the full literature. It would be really interesting to see how the distributions provided by NOFLITE's use of Normalizing Flows would compare to those generated by the Neural ODE based CF-ODE model. A comparative analysis that could be done is in measuring each distribution's calibration for each possible treatment (see Sun and Yu, 2022 for various calibration metrics).

It's unclear why it's necessary to have a monotonicity requirement in the Normalizing Flow. It would be helpful to more clearly lay out the assumptions and constraints used in the formulation of NOFLITE.

It's unclear whether comparing the empirical baselines to NOFLITE via the model's likelihood is totally fair given that the baselines are not optimized to maximize a likelihood? Some clarification from the authors in the text on this point, justifying the use of this comparative metric would be really helpful.

More discussion should be included at the beginning of Section 3 to illustrate/explain why the expectation is "insufficient for decision-making in may domains". It's not appropriate to vaguely present these types of arguments without more information.

The major and possible disqualifying weaknesses, such that the major claims of the paper are unsupported are:
**There is no convincing demonstration that supports the major claim of the paper, that a full treatment effect distribution is necessary for adequate decision making. The empirical analysis rests on three metrics and does not discuss the implications of downstream use. On top of this, the improvements seen by NOFLITE are marginal at best and are not consistent across all metrics.**

**There is zero demonstration or analysis to support the suggested strengths of the full distribution, more coherent statistical analysis _and_ the use of "personalized utility functions".** <-- This is a major disappointment because the use of utility functions is a neat idea and I was disappointed that there was no supporting evidence for the concepts and ideas put forward at the beginning of the paper.

##### Suggested Literature
- `Edward De Brouwer, Javier Gonzalez, and Stephanie Hyland. Predicting the impact of treatments over time with uncertainty aware neural differential equations. AISTATS 2022`
- `Sophia Sun and Rose Yu. Copula conformal prediction for multi-step time series forecasting. arXiv preprint arXiv:2212.03281, 2022`

---

> ### Author Response · Authors · 2023-10-06
> **Response to Reviewer KX1e: Part 1/3**
>
> Dear Reviewer KX1e,
>
> Thank you  for your comments and for dedicating time to reviewing our work. In the following, we address each of the major points raised in your review: (1) Motivation of our work, (2) Evaluation metrics, and (3) Baselines. We end with a response to several smaller remarks in (4) Minor points.
> ___
>
> # (1) Motivation of our work
> The reviewer asks us to clarify the *"the actual claims and intended contributions of this paper"*. We explicitly state the contributions of our work in the introduction, but feel like the confusion stems from the motivation of our method. Therefore, we will expand upon two key points: (a) the (expected) utility and (b) statistics derived from the individual treatment effect distribution. We hope that this more extensive explanation will clarify the research problem our work aims to address.
>
> ## (a) Utility functions
> First of all, the reviewer asks to motivate *"why utility functions are important or useful"* and to explain *"why expected utility is a desirable metric"*. We will first answer these questions in general and then explain in more detail how this relates to our work.
>
> In general, a utility function describes the utility that an indivual derives from a variable of interest. It is an essential concept required for decision-making under uncertainty, which has been important in a wide range of fields (see the references in the introduction).
>
> To make this concept more clear, we give an example and consider a seminal use case of utility functions. In economics, an agent's utility of money $u(y)$ describes the value they would derive from an amount of money $y$ (e.g., their happiness level given an amount of money). In this case, a person that is risk averse would have a utility function that is concave, e.g., $u(y) = \sqrt{y}$. Imagine that a risk averse person with $u(y) = \sqrt{y}$ has $\\$10$ and considers buying a lottery ticket. The ticket costs $\\$2$ and has an uncertain payoff: $ \\$200$ with probability $1\%$, zero otherwise. The **expected payoff** of buying this lottery ticket is $0=-2 + 200 \times 0.01 + 0 \times 0.99$ Nevertheless, this person is *not* indifferent between buying the ticket and not buying it, as their **expected utility** of buying the ticket is negative: $$E[U(y|\text{Ticket})] - U(y|\text{No ticket}) \\ = \big( 0.01 \times U(10 - 2 + 200) + 0.99 \times U(10-2) \big) - U(10) \\ = 0.01 \times \sqrt{208} + 0.99 \times \sqrt{10-2} - \sqrt{10} \approx \\ = -0.22$$.
>
> This simple example illustrates a key motivation of our method: *the expectation of the outcome is not sufficient for decision-making under uncertainty*. Indeed, assuming that this person's preference could be based on the lottery ticket's expected value by itself implicitly assumes a linear utility function $U(x)=ax+b$.
>
> Our work is motivated by the many applications in which we wish to *act* upon an outcome of interest with an action, intervention, or treatment, where the change in outcome resulting from that treatment is unknown and needs to be estimated from data. In this setting, the utility function can be used to help decide on treating or not, or to compare different treatments. In our example, the treatment could be an educational program with the aim being to increase that individual's earnings. Importantly, in contrast to the lottery ticket, we do not know how this treatment will affect the outcome, but have to estimate this from data. Existing work in causal inference is typically limited to estimating the expected value of this change, which is not sufficient—as illustrated by the example above. To accommodate a utility function, we require knowing the distribution of the treatment effect.
>
> In conclusion, the utility function has the potential to be a useful tool in many applications where causal inference is currently being used. However, to use a utility function for decision-making under uncertainty, we need to pair it with the individual treatment effect distribution. Unfortunately, most existing work does not focus on predicting this distribution, but only the expected effect—this is a research gap we aim to address.

---

> > ### Comment · Reviewer_KX1e · 2023-10-08
> > **Contributions vs. claims**
> >
> > I have stated that I feel that the contributions of the paper are strong and interesting. However, technical novelty as recorded contributions are not the same as claims which could be better construed as hypotheses or points of interrogation following the introduction of new methods, datasets and/or approaches. **The most generous I could be on this point would be to accept the implicitly stated assumption (echoing my full review, where this is not precisely put forward by the authors) that estimating the full treatment effect distribution leads to better decision making under uncertainty via the enabling of utility functions and more descriptive statistics.**
> >
> > If these are the claims the authors intend to make, great! If so, then the experiments should be used to support and verify these claims. As currently written, I do not believe that the entirety of the claims are supported by the experiments. From the author's response, I am convinced to a degree that the claims about the use of descriptive statistics are supported. **However there are no experiments verifying the use of utility functions or demonstrating how they would affect the performance.**
> >
> > The focus of the author's response appears to argue that the development of the treatment effect distribution alone is sufficient to support the hypothetical need or use of a utility function. The argument seems to center on an appeal for generality since specific utilities may be different under separate uses/dataset/etc with NOFLITE. However, based on the claims in the paper and no true demonstration of what this may entail empirically (outside of the hypothetical example in Figure 1) this is far from sufficient in my view.
> >
> > Now, an alternative argument that seems to be made by the authors is that the intention of the experiments is to demonstrate that NOFLITE learn better distributions. This is fine, and appears to be well supported by the experimental results, but appears to be a separate empirical hypothesis from what the paper has put forward in view of the implicit claims. This is a little confusing when the notion of decision making is used as motivation, which demands a little more specificity with regards to the formulation and results. I hope the authors can see why I have had a difficult time clearly identifying what the intended claims of the work to be.
> >
> > Clarifying these aspects of the work within the paper would greatly improve it and help me to better advocate for its publication. I will make separate responses to other points of contention where additional clarification is or is not needed.

---

> > > ### Author Response · Authors · 2023-10-13
> > > **Response to "Contributions vs. claims" and "Re: evaluation metric"**
> > >
> > > Thank you once again for your detailed and swift response! Below, we jointly respond to your replies "Contributions vs. claims" and "Re: evaluation metric".
> > > ___
> > > We believe that we now understand your point regarding our claims. To address this issue and better align our motivation with our experiments, we have (1) added a preface to our results section to more clearly state the aims of our experiments, and (2) included additional results using utility functions for decision-making.
> > >
> > > For the preface to our results section, we now clearly state the aims of our experiments and the related evaluation metrics:
> > > "More specifically, our experiments aim to answer three questions. We give an overview of these questions and related metrics here; more detail is provided below. *(1) Does NOFLITE predict accurate individual treatment effect distributions?* This is our main question of interest. We evaluate this with the loglikelihood. *(2) Does the predicted distribution allow for a more detailed analysis of individual the treatment effect, based on statistics derived from this distribution?* This is evaluated by looking at the predicted distribution's expected value (using the precision in estimation of heterogeneous treatment effects), as well as the derived confidence intervals (using the intersection-over-union and empirical coverage). *(3) Does the predicted distribution enable accurate decision-making when paired with a utility function?* We evaluate this by looking at the accuracy of the recommended treatments."
> > >
> > > We believe that this also addresses your point with respect to our use of the likelihood metric, by clarifying that this aims to answer question (1).
> > >
> > > We agree that there were *"no experiments verifying the use of utility functions"*, while this was mentioned as a motivation for our method. Therefore, we now explicitly add this point as question (3) and have added additional results to address this. To this aim, we have paired the predicted distributions with a given utility function, and assessed whether the expected utility could be used for decision-making. For IHDP, we use the following utility function: $u(\tau) = (\tau - 4)^3$, given that the average treatment effect will be $4$ on average (based on the data generating process). For EDU, we use $u(\tau) = (\tau - 1)^3$, as the treatment effect was empirically observed to be approximately $1$ on average. The predicted optimal treatment decision is compared with the theoretical optimal based on the ground truth distribution using the accuracy. Note that we cannot do this analysis for the News data, as the true distribution is not known.
> > >
> > > We show the empirical results below. Note that NOFLITE's distributions result in relatively good decisions. In general, we can see that models with a higher likelihood also achieve better decision quality.
> > >
> > > #### Additional results: Decision accuracy given $u(\tau)$
> > > |              | IHDP         | EDU          |
> > > | ------------ | ------------ | ------------ |
> > > | **CMGP**     | 0.90         | 0.70         |
> > > | **BART**     | 0.84         | 0.69         |
> > > | **GANITE**   | 0.55         | 0.72         |
> > > | **CEVAE**    | 0.57         | 0.49         |
> > > | **CCN**      | 0.86         | 0.76         |
> > > | **NOFLITE**  | 0.90         | 0.76         |
> > >
> > > We consider these results a valuable addition to our work and believe that our motivation, contributions, and claims are now more clear. Thank you for these suggestions!

---

> ### Author Response · Authors · 2023-10-06
> **Response to Reviewer KX1e: Part 2/3**
>
> ## (b) Statistics derived from the distribution
>
> Regarding the second motivation for our method, the reviewer similarly asks: *"What do the expanded set of descriptive statistics actually provide?"* The individual treatment effect distribution and statistics derived from this distribution could be benefial for decision-makers, without engaging in a formal analysis of the utility. As an example, consider a patient considering a treatment that will affect their remaining life years by an uncertain effect $\tau$. Even though the expected effect $E(\tau)$ may be positive, the patient may want to know the probability of the treatment having a detrimental effect $p(\tau < 0)$. Similarly, a long left tail may indicate that there is a high risk of adverse effects. Finally, a confidence interval may give the patient a reasonable expectation of the effect. Although this is just one example of an application, we hope that this type of reasoning is relatble and illustrates the importance of having these statistics when making a decision under uncertainty.
>
> Estimating a distribution to have access to these additional statistics is something that has been mostly overlooked in existing work, which focused on the expected value only. Again, this is a research gap we aim to address.
>
> Finally, we stress that neither the utility function nor the statistics derived from this distribution aim to *"modify and/or improve the analysis of the estimated treatment effect distributions"*, as suggested by the reviewer. Estimating of the ITE distribution precedes the calculation of the expected utility or the analysis of specific statistics derived from this distribution. In turn, these can be used for decision-making. In other words, we do not make assumptions regarding the decision-making process that will be used—in terms of specific utility functions or statistics. Rather, we aim to provide the individual treatment effect distribution as a tool for decision-makers. Moreover, we remain agnostic with respect to the actual utility functions or statistics used in the decision-making process. Rather, we choose to focus on estimating the treatment effect distribution itself, a necessary requirement for these analyses.
>
> We hope that this more substantial explanation clarifies the importance of the problem that our work addresses. We would be more than happy to include a more in-depth discussion in our work if the reviewer believes that it makes our motivation more clear.
>
>
> ___
>
> # (2) Evaluation metrics
> The reviewers asks us to *"clarify how the log likelihood is an adequate comparative metric across the various baselines"*. Given a model $\theta$, the (log)likehood denotes the probability of the observed data for that model $p(\tau | \theta)$. This way, the likelihood allows for assessing how well the predicted distributions fit the empirical data. This metric is the standard approach for evaluating models that output a distribution in machine learning, e.g. in the context of generative models (Theis et al., 2015). For our case specifically, the loglikelihood can be used to objectively compare different baselines because we compute it an a model-independent way: by fitting a kernel density estimator on each model's estimated samples, the likelihood calculation is the same for each model. Finally, the fact that NOFLITE can be optimized based on the likelihood directly is a specific motivation for our method, which distinguishes it from other methods such as CEVAE and GANITE.
>
> The loglikelihood gives a measure of the *global* quality of the distribution. However, we can also evaluate specific statistics derived from this distribution. In our work, we presented results for quality of the estimated expected values $E(\tau_i)$ in terms of PEHE and 90\% confidence intervals $CI$ in terms of IoU. In the updated manuscript, we additionally added empirical coverage results to evaluate the confidence interval. Finally, we also add qualitative results in Figure 3, showcasing NOFLITE’s output for several instances.
>
> There are many other statistics that could be evaluated. However, we would like to stress again that our work is primarily concerned in the loglikelihood as a metric for evaluating the quality of the predicted distribution. Because of this, we do not agree with the reviewer that *"the improvements seen by NOFLITE are marginal at best and are not consistent across all metrics"*. Our method has consistent strong performance across data sets in terms of loglikelihood -- our main metric of interest. As we discuss in the results section, other methods may be preferable if the goal is to obtain good performance for a single, specific metric. For example, methods optimizing the MSE may get more accurate expected values, or methods for quantile regression might get accurate confidence intervals. In contrast, NOFLITE performs well when we are interested in predicting the distribution, which motivated our work.

---

> > ### Comment · Reviewer_KX1e · 2023-10-08
> > **Re: evaluation metric**
> >
> > In light of the intended focus of "estimating the treatment distribution itself", the use of the log-likelihood as a metric is obviously appropriate. However, when a subset of the methods are optimized using the log-likelihood while others are not (or eschew estimating the distribution at all), that is where I am concerned about the fairness of comparison. My concern lies in the inconsistency of where the comparisons are drawn (estimating the distribution vs. make accurate inferences). I acknowledge that this can be a tricky balance to make given the prevalence of recent generative approaches to estimating treatment effects along the lines of the hypothesis and intended contributions of NOFLITE. Perhaps a bit more clarity in the experimental setup section around what the intended insights are by making the comparisons of NOFLITE to the baseline methods would help?

---

> ### Author Response · Authors · 2023-10-06
> **Response to Reviewer KX1e: Part 3/3**
>
> # (3) Baselines
>
> The reviewer asks to include CF-ODE as an empirical baseline. We would be more than happy to include more benchmarks (see the inclusion of BART based on Reviewer gFHo's suggestion). However, CF-ODE aims to estimate treatment outcomes *over time*. In contrast, our work focuses on a *static setting*. Therefore, we do not see how CF-ODE is applicable as a baseline in our problem setting. Similarly, the paper by Sun and Yu (2022) is focused on time series prediction. We do not see how this can be applied in our setting.
>
>
> ___
>
>
> # (4) Minor points
>
> We discuss several minor points in your review in the following: (a) Details on normalizing flows, (b) Figure 2, and (3) Clarification on metalearners.
>
> ## (a) Details on normalizing flows
> The reviewer asks to provide more detail about how the Normalizing Flows are used in practice. In general, any sequence of Normalizing Flow transformations $\mathbf{g}$ can be used within our framework. Each transformation $g$ needs to be invertible, as this allows to calculate (and optimizing) the likelihood exactly. As we believe a complete introduction of normalizing flows is outside of the scope of our work, we refer to the excellent survey by Papamakarios et al. (2021) for a more extensive introduction.
>
> Within the NOFLITE framework, the encoder first predicts an instance's normal distribution by predicting two parameters $\mu$ and $\sigma$. Normalizing flows are then used to transform this simple distribution to a more complex posterior that matches the empirical distribution $p(y|x,t)$. In our work, we use deep sigmoidal flows (Huang et al., 2018). Deep sigmoidal flows achieve the invertibility requirement of the mapping by defining the flow transformation as a strictly monotonic transformation. More specifically, the transformation is a monotonic neural network, whose parameters are estimated by a second conditioner network. By constraining the outputs of this conditioner network, the monotonicity of the transformer network is ensured. Specifically, the transformer network has strictly positive weights and strictly monotonic activation functions. Although the resulting mapping is invertible, it is not known analytically. Therefore, we make use of an interpolation search to find it numerically.
>
> We would also like to point out that all our code will be made publicly available, including our implementation of NOFLITE.
>
> Thank you for pointing out the missing inversion $g^{-1}$ in Figure 2! We have updated this in the revised manuscript.
>
>
> ## (b) Figure 2
>
> Regarding Figure 2, the reviewer states that *"More detail is warranted there to explain the balanced representations used."* Although we are happy to adjust our figures, we feel like adding this would be confusing. This is because, while we predict treatment effect distribution *for each instance*, the representations are balanced *over instances*. Therefore, we are afraid that adding a graphic of the distribution over treatment and control group would be confusing. If the reviewer has any specific ideas on how to represent this, we would be more than happy to include them. Finally, we also provide details on the training of our method in Section 4.2 optimization, including a discussion of how we learn the balanced representation.
>
> ## \(c\) Section 4.1.2: Clarification on metalearners
>
> The reviewer argues that *"Section 4.1.3 [Metalearner configuration] seems unnecessary in the main body of the paper without supporting context in Section 2 [Normalizing flows]."*
>
> Thank you for this suggestion! We have added a reference and discussion on metalearners in the related work section. Nevertheless, if the reviewer believes that additionally moving Section 4.1.3 to the appendix makes our work more streamlined, we would be happy to do so.
>
>
> ___
>
>
> Again, we would like to thank the reviewer for their suggestions and time spent on reviewing our work. While we hope that these responses address your concerns, please let us know if there are any outstanding issues—we would be more than happy to engage in further discussion.
>
> ___
>
> ### _References_
> * Papamakarios, G., Nalisnick, E., Rezende, D. J., Mohamed, S., & Lakshminarayanan, B. (2021). Normalizing flows for probabilistic modeling and inference. The Journal of Machine Learning Research, 22(1), 2617-2680.
> * Huang, Chin-Wei, David Krueger, Alexandre Lacoste, and Aaron Courville. "Neural autoregressive flows." In International Conference on Machine Learning, pp. 2078-2087. PMLR, 2018.
> * Theis, L., Oord, A. V. D., & Bethge, M. (2015). A note on the evaluation of generative models. arXiv preprint arXiv:1511.01844.

---

> > ### Comment · Reviewer_KX1e · 2023-10-08
> > **Continuing thoughts**
> >
> > **Re: baselines**
> >
> > I understand that the authors have deemed CF-ODE to be out of scope as an empirical baseline. I do however feel that it is important to be included in the related work as it is a contemporary approach that addresses many of the same points that underlie the development of NOFLITE.
> >
> > Sun and Yu (2022) was not suggested as a baseline but a suggestion for ideas of how the inferred distributions between competing approaches could possibly be evaluated.
> >
> > **Normalizing flows**
> >
> > It was not my intention to recommend that the authors exhaustively introduce the concepts underlying Normalizing Flows. It was however my intention to encourage the authors to be a bit more direct and clear about the mechanisms used in NOFLITE. The description given here is much better detailed than what was in the original submission. I would recommend that it be incorporated into Section 4.
> >
> > **Meta-learners**
> >
> > Thanks for adding these parts to the related work. I retract my recommendation that Section 4.1.3 is unnecessary. I took another read through the paper and it should stay as part of the main body of the paper. Forgive my lack of insight.

---

> > > ### Author Response · Authors · 2023-10-13
> > > **Response to "Continuing thoughts"**
> > >
> > > Below, we provided responses to "Re: baselines" and "Normalizing flows".
> > >
> > > ## Re: baselines
> > >
> > > We agree that, although CF-ODE considers a different setting, it is an interesting related work. We have added the reference to our related work section.
> > >
> > > ## Section 4
> > >
> > > We have added our additional explanation on normalizing flows in NOFLITE to section 4.
> > >
> > > ___
> > >
> > > Thank you once again for your time and effort in reviewing our work—this is highly appreciated! We believe that our discussion improved the quality of our work and hope that these changes adequately addressed your concerns. However, please do let us know if there are any remaining issues.

---

> ### Comment · Reviewer_KX1e · 2023-10-19
> **A final nitpicking comment**
>
> Thank you for taking the time to evaluate these additional experiments. I do feel that it's helped round out the empirical justification of the claims. What's not overly clear is whether you apply the same utility function to all of the baseline methods? Is that well supported as they do not generate a treatment effect distribution? I suppose that what I was expecting to see was a comparison between nominal predictions (using the expected value) for all methods (including NOFLITE) and then having an additional set of rows for NOFLITE + various utility functions.
>
> I understand that the chosen utility functions for IHDP and EDU were determined based on a priori knowledge of the generating distribution. This does feel a little bit of a hack though. I think that the strength of the this comparison could be through the use of  different forms of utility function (risk seeking, risk neutral, risk averse) and then identify what may be best suited for each dataset with a commentary on how these specific utility functions improve NOFLITE performance above and beyond the baselines. This is possibly what would be expected if you were analyzing a new dataset in practice anyway? You'd validate the choice of utility function on a held out portion of the data before performing your final evaluation? In this way, you may be able to recover the comparisons for the News data. Also, looking at the most recent revision of the paper, it appears that these details about what specific utility functions are used are missing. Please be sure to be more direct and clear about what the experimental comparisons are actually measuring!
>
> Additionally, I'm interested to see if the authors could provide confidence intervals for these scores? That would help differentiate the performance of NOFLITE against closely matched baselines (CMGP for IHDP and CCN for EDU)...

---

> > ### Author Response · Authors · 2023-10-20
> > **Response to: "A final nitpicking comment"**
> >
> > Thank you for your response!
> >
> > Indeed, we apply the same utility function for all methods, as the goal is to see whether decisions are optimal *for a given utility function*. All methods included as benchmarks can in fact predict treatment effect distributions — this is exactly why they were chosen as benchmarks. Because of this, all methods can be paired with a utility function in the same way, enabling a fair comparison across methods. Conversely, we do not feel that comparing decisions based on nominal values would offer a fair comparison, as these *cannot be optimal for any method* (or even when using the ground truth).
> >
> > With respect to the choice of utility function, we made this based on two criteria. First, the $u(\tau) = x^3$ was chosen to be a non-linear function, as the expected value itself might not be sufficient in this case. Second, we use a constant to make sure that it is not optimal to always recommend treatment or to never recommend treatment. Although this is indeed based on knowledge of the data generating process, we would argue that it makes for a more interesting decision-making problem and that it should not impact different methods in different ways.
> >
> > In general, all kinds of utility functions are possible. Choosing what utility function is applicable in practice should be done by decision-makers with expertise in that particular domain. Note that these functions may even be different for each instance (e.g., different patients may have different risk profiles). Moreover, it is not possible to compare results across utility functions, as different decisions could be optimal for different utility functions. Because of this, we cannot *"validate the choice of utility function on a held out portion of the data"*, but require the utility function to be specified a priori.
> >
> > What we believe is important is that our results show that **models with high likelihood also result in good decisions** — which motivated our method. Therefore, we believe that these results support our contribution: a method that is capable of learning accurate individual treatment effect distributions, which has useful implications for decision-making. The result with the utility function serves as an illustration of this point. However, our main focus remains predicting accurate distributions.
> >
> > With respect to the confidence intervals: we provided standard errors for all results in the updated draft of the paper. For your convenience, we have also added them in the table below. The confidence intervals are relatively small for all methods.
> >
> > #### Additional results: Decision accuracy given $u(\tau)$
> > |              | IHDP         | EDU          |
> > | ------------ | ------------ | ------------ |
> > | **CMGP**     | 0.90 $\pm$ 0.01       | 0.70 $\pm$ 0.00       |
> > | **BART**     | 0.84 $\pm$ 0.01       | 0.69 $\pm$ 0.00       |
> > | **GANITE**   | 0.55 $\pm$ 0.02       | 0.72 $\pm$ 0.01       |
> > | **CEVAE**    | 0.57 $\pm$ 0.01       | 0.49 $\pm$ 0.01       |
> > | **CCN**      | 0.86 $\pm$ 0.01       | 0.76 $\pm$ 0.01       |
> > | **NOFLITE**  | 0.90 $\pm$ 0.01       | 0.76 $\pm$ 0.01       |
> > ___
> > We hope that this additional discussion clarifies our design choices and experimental setup. Thank you once again!

---

### Review · Reviewer_gFHo · 2023-09-19

**Summary Of Contributions:**

This paper proposes a neural network based model for estimating individual treatment effect distributions. The authors approach this problem by using two headed network, using normalizing flows in order to allow for more complex distributions, and imposing a balance constraint using maximum mean discrepancy in order to have validity of the estimated counterfactuals. The core of the idea is to combine probabilistic modeling with normalizing flows and ideas from estimation of the mean potential outcomes using neural networks. Experiments are provided which demonstrate the efficacy of the proposed method.

**Audience:**

Yes

**Broader Impact Concerns:**

Individual effect estimation has immediate impact to optimization and recommendation algorithms that various actors may use to alter of manipulate the behavior of people and other systems. While this work does not provide any immediate ethical quandaries past what currently exist in the prior art it is important to be mindful of.

**Claims And Evidence:**

Yes

**Requested Changes:**

As I mention above, I would like to see a stronger motivation for the balance constraint from a probabilistic perspective, an extended set of simulation results that compare against more "classical" benchmarks (which remain very strong and competitive baselines). It would also be nice if the authors discussed theoretical properties along the lines of Johannson, et al. (Learning Weighted Representations for Generalization Across Designs), though I understand if the authors feel this is beyond the scope of the paper.

**Strengths And Weaknesses:**

Strengths:
* This is a simple (in a good way) idea to combine ideas from generative modeling and the literature on causal neural networks to obtain distribution estimates.
* The performance of the proposed estimator is quite good compared to other neural network based approaches.
* Well written (very nice clarity in both task description and solution) and organized.
* I can see how the proposed method would be easily extensible and could provide a nice basis for future work.

Weaknesses:
* The inclusion of the MMD constraint in this setting is motivated as it is in the prior literature (to enforce balance). Given the probabilistic framing for the rest of the paper it could be useful to have a larger discussion about how the MMD constraint should be interpreted in the probabilistic context (e.g., the discussion in Kanagawa, et al. proposition 6.1 (https://arxiv.org/pdf/1807.02582.pdf).
* It would be useful to have a comparison against non-neural network based estimators as well, especially BART which is mentioned in related work but not compared to.
* The paper has no theoretical results. While this isn't necessarily a reason to prevent acceptance it would be nice if there were some discussion on the properties of the proposed estimator.

---

> ### Author Response · Authors · 2023-10-06
> **Response to Reviewer gFHo**
>
> Dear Reviewer gFHo,
>
> Thank you for your insightful review of our work and your thoughtful comments and suggestions! We are delighted by your positive feedback, appreciating clarity of our writing and conceptual simplicity of our method. We agree with the reviewer and hope that NOFLITE can provide the starting point for more work in this area.
>
> Below, we give point-by-point responses to the major open points raised in your review: (1) Motivating the balance constraint, (2) Comparing against more classical benchmarks, and (3) Theoretical properties.
>
> ***
>
> # (1) Motivating the balance constraint
>
> While we agree with the reviewer that the analysis in Kanagawa et al. (2019) that interprets the balance constraint from a probabilistic persective is interesting, we are not sure that their reasoning can be applied to our methodology. We balance the representation at the level of all control and test instances, as the MMD is calculated over (a batch of) instances. Conversely, NOFLITE predicts distributions for one particular instance. Therefore, the probabilistic perspective would be different from the perspective NOFLITE takes in general.
>
> Moreover, we believe that our specific contribution is predicting the treatment effect distribution. To this aim, we mostly use balancing to deal with confounding bias based on its state-of-the-art performance in previous work. However, other methods could also be used with our framework (e.g., inverse propensity score weighting).
>
> ___
>
> # (2) Comparing against more classical benchmarks
>
> We agree with the reviewer that comparing against other non-neural approaches could offer valuable insights into the performance of our method. In the original submission, we already included results of CMGP, a benchmark based on Gaussian processes. Regarding BART, we agree that this is an interesting benchmark -- thank you for this suggestion! We have added results for BART below and in the updated manuscript.
>
>
> #### **Empirical results: comparing BART and NOFLITE**
> |             | **IHDP** |        |       |       | **EDU** |        |       |       | **News** |        |        |
> |-------------|----------|--------|-------|-------|---------|--------|-------|-------|----------|--------|--------|
> |             | _LL_     | _PEHE_ | _IoU_ | _Cov_ | _LL_    | _PEHE_ | _IoU_ | _Cov_ | _LL_     | _PEHE_ | _Cov_  |
> | **BART**    | -2.16    | 2.22   | 0.63  | 0.83  | -1.71   | 0.53   | 0.56  | 0.89  | -2.43    | 2.71   | 0.97   |
> | **NOFLITE** | -1.90    | 1.09   | 0.75  | 0.88  | -1.62   | 0.26   | 0.64  | 0.89  | -2.15    | 2.18   | 0.93   |
>
> Although BART performs relatively well compared the other benchmarks, NOFLITE achieves better results. For the complete experimental results, we refer to the updated version of our paper. Note that these include empirical coverage results based on the feedback from Reviewer 6esJ.
>
> # (3) Theoretical properties
>
> While we agree with the reviewer that analyzing the theoretical properties of NOFLITE would be insightful, we indeed consider this out of the scope of *the current work. Nevertheless, we also believe that this constitutes a fruitful area for future work. To encourage future research in this area, we have updated the conclusion to explicitly include this as a topic for future work.
>
> ___
>
> We would once more like to thank the reviewer for their time and constructive feedback! We hope the above adequately addresses your comments. Nevertheless, if there are any remaining questions, we would be more than happy to engage in further discussion.

---

### Decision · Action_Editor_rhMu · 2023-10-27

**Recommendation:** Accept with minor revision

**Comment:**

The paper studies a problem relevant to the TMLR community and approaches it from a new perspective. After the substantial revision and discussion between authors and reviewers, all reviewers leaned toward accepting the paper.

However, the main motivating idea—that the average effect is not enough for good decision making—has many more links to previous work than acknowledged in the related work section. For the final version of the paper, I recommend adding a paragraph, or substantially updating the second paragraph in related works, contextualizing the manuscript in the closely related literature on reinforcement learning, off-policy evaluation, and conformal prediction of treatment effects. To name a few *recent* examples,
* Wang, L., Zhou, Y., Song, R., & Sherwood, B. (2018). Quantile-optimal treatment regimes. Journal of the American Statistical Association, 113(523), 1243-1254.
* Chandak, Y., Niekum, S., da Silva, B., Learned-Miller, E., Brunskill, E., & Thomas, P. S. (2021). Universal off-policy evaluation. Advances in Neural Information Processing Systems, 34, 27475-27490.
* Bellemare, M. G., Dabney, W., & Rowland, M. (2023). Distributional reinforcement learning. MIT Press.
* Xu, T., Yang, Z., Wang, Z., & Liang, Y. (2022). A Unifying Framework of Off-Policy General Value Function Evaluation. Advances in Neural Information Processing Systems, 35, 13570-13583.
* Wu, R., Uehara, M., & Sun, W. (2023). Distributional Offline Policy Evaluation with Predictive Error Guarantees. arXiv preprint arXiv:2302.09456.
* Zhang, Y., Shi, C., & Luo, S. (2023, April). Conformal Off-Policy Prediction. In International Conference on Artificial Intelligence and Statistics (pp. 2751-2768). PMLR.

I am confident that the authors can find even more examples of aiming to estimate the full distribution of treatment effects / policy rewards / utility functions.

**Audience:**

Yes.

**Claims And Evidence:**

Following the revisions recommended by reviewers, the paper supports its claims with sufficient and clear evidence.

---

> ### Author Response · Authors · 2023-11-09
>
> Dear action editor, dear reviewers,
>
>
> We have uploaded a camera-ready revision of our paper, where we have added a paragraph of related work with a similar motivation to ours, i.e., that the average effect is insufficient for decision-making.
>
> We would like to express our gratitude to everyone involved for their time, effort, and valuable discussions throughout this review process—thank you!
>
>
> Best regards,
>
> The authors